# Pleiotropic Genes Affecting Milk Production, Fertility, and Health in Thai-Holstein Crossbred Dairy Cattle: A GWAS Approach

**DOI:** 10.3390/ani15091320

**Published:** 2025-05-02

**Authors:** Akhmad Fathoni, Wuttigrai Boonkum, Vibuntita Chankitisakul, Sayan Buaban, Monchai Duangjinda

**Affiliations:** 1Department of Animal Science, Faculty of Agriculture, Khon Kaen University, Khon Kaen 40002, Thailand; akhmad.f@kkumail.com (A.F.); wuttbo@kku.ac.th (W.B.); vibuch@kku.ac.th (V.C.); 2Department of Animal Breeding and Reproduction, Faculty of Animal Science, Universitas Gadjah Mada, Yogyakarta 55281, Indonesia; 3Network Center for Animal Breeding and Omics Research, Faculty of Agriculture, Khon Kaen University, Khon Kaen 40002, Thailand; 4Bureau of Animal Husbandry and Genetic Improvement, Department of Livestock Development, Pathum Thani 12000, Thailand; buaban_ai@hotmail.com

**Keywords:** GWAS, pleiotropic genes, genomic prediction, wssGBLUP, milk yield, fat-to-protein ratio, days open, dairy cattle breeding

## Abstract

Genomic selection and genome-wide association studies (GWASs) are powerful tools for improving dairy cattle’s milk production, fertility, and health traits. While these methods have been successfully applied to purebred populations in the United States and Europe, their effectiveness in crossbred dairy cattle, particularly in tropical regions such as Thailand, remains less well explored. Identifying the pleiotropic genes influencing multiple economically important traits can enhance genetic selection strategies. This study evaluated the potential of weighted single-step genomic best linear unbiased prediction (wssGBLUP) to improve GEBV accuracy compared to the standard single-step GBLUP (ssGBLUP) and applied a wssGWAS to identify key genetic markers for production (305-day milk yield), fertility (days open), and health traits (fat-to-protein ratio) in Thai-Holstein crossbred dairy cattle. Further research is needed to validate these findings and optimize breeding programs for tropical dairy systems.

## 1. Introduction

Sustainable dairy production in tropical regions necessitates cattle breeds with both high milk yield and heat tolerance. Thai-Holstein crossbred cattle, a product of a breeding program initiated in 1956 [1], represent a crucial resource for Thailand’s dairy industry, combining indigenous breeds’ resilience with Holsteins’ superior milk production. Notably, over 95% of these crossbreds result from crosses between Holstein and either the Sahiwal or Thai Native breeds. The crossing of these breeds was intended to increase milk production from taurine cattle, while incorporating tolerance of heat stress, ticks, and tropical diseases. However, a historical focus on maximizing milk yield has inadvertently compromised fertility and overall animal health [2,3,4], highlighting the need for a more integrated breeding approach.

Negative energy balance (NEB), a key factor in the compromises between milk yield, fertility, and health, arises when energy intake fails to meet the high metabolic demands of early lactation [5,6]. During early lactation, if the energy intake of a cow is insufficient to meet metabolic demands, it enters a state of NEB. In severe cases, NEB leads to metabolic stress, reduced feed intake, higher disease susceptibility, metabolic disorders, and impaired reproductive performance [5,7]. These conditions are associated with metabolic diseases such as ketosis and fatty liver syndrome [5,7], where cows mobilize body fat to compensate for energy deficits, resulting in increased ketone production and impaired liver function [8,9]. Additionally, a deficiency in fermentable carbohydrates reduces protein synthesis by rumen bacteria, leading to lower amino acid flow to the udder, decreased milk protein, and an elevated milk fat-to-protein ratio [10,11].

The reproductive consequences of NEB include delayed ovulation, reduced conception rates, and extended calving intervals [12,13], often driven by hormonal imbalances that disrupt the estrous cycle and follicular development [14]. Similarly, an excessively low milk FPR can indicate subclinical rumen acidosis, a metabolic disorder characterized by a sharp drop in rumen pH. This condition can lead to severe health complications [15], including laminitis and immune suppression, increasing susceptibility to infections [16]. Therefore, the optimal milk FPR in tropical Holsteins ranges from 0.9 to 1.9 [8], emphasizing its value as an indicator of overall cow health and productivity.

Milk FPR serves as a potential indicator of NEB and exhibits significant associations with milk yield and fertility traits, demonstrating heritability ranging from 0.19 to 0.54 across various dairy breeds [11,17,18]. Genetic correlations between milk yield and milk FPR vary across multiple breeds, showing positive correlations in some (e.g., Nordic Red and Canadian Holstein: 0.05–0.48 in early lactation [18,19]) but negative correlations in others (e.g., Thai dairy cattle: −0.44 to −0.29 in late lactation [18]). Similarly, correlations between milk FPR and DO range from 0.37 to 0.67 to 0.98 in Thai dairy cattle and from 0.14 to 0.28 in Nordic Red cattle [19], suggesting a compromise between energy balance, milk production, and reproductive performance. Previous studies have primarily focused on phenotypic traits or common genetic variations [20,21,22,23,24,25,26]. Given the importance of genomics selection for these traits [27,28], this study aimed to identify pleiotropic genes associated with MY305, DO, and milk FPR using a GWAS and to analyze their relationships using multi-trait ssGBLUP models.

## 2. Materials and Methods

### 2.1. Data Collection

This study examined 18,843 records of MY305 and FPR from first lactation and 48,274 records of DO after first calving in Thai-Holstein crossbred dairy cows collected between 1993 and 2017. The data were provided by the Bureau of Biotechnology in Livestock Production, Department of Livestock Development, Thailand. AI data were recorded on farms that used the DLD’s bulls, while test-day milk and milk composition data were collected only from selected farms. MY305 denotes the total milk production (kg) during the first 305 days of lactation, calculated from test-day milk data. After merging the data, 897 farms, with an average herd size of 54.0 cows/farm, were used in the study. DO was defined as the number of days between calving and conception during the lactation period. A conception record was discarded if the consecutive calving date was unknown. Cows with a DO value less than 20 or greater than 400 were also removed. The calving age was restricted to between 24 and 48 months. Milk FPR was defined as the ratio of milk fat to protein in test-day milk samples, with the average value over the lactation period being used for analysis. Cows were grouped by percentage of Holstein genetics (breeding group, BG) as follows: BG1 > 93.7%, BG2 from 87.5 to 93.6%, and BG3 < 87.5%.

Genotype data were obtained from 882 Thai-Holstein crossbred cows using the Illumina BovineSNP50 Bead Chip (Illumina Inc., San Diego, CA, USA). Genotyped animals were randomly selected based on high, medium, and low breeding values of bulls and dams. Quality control (QC) was applied to both animals and markers based on the following criteria: minor allele frequency (MAF) > 5% and minimum call rate per marker ≥ 90%. The SNPs and animals that did not meet these criteria were excluded. Animals with duplicate genotypes and SNPs with Mendelian conflicts were also discarded. Quality control and SNP analyses were performed using the BLUPF90+ Ver 2.56 software. After QC, 868 genotyped animals and 43,284 informative SNPs for MY305, DO, and milk FPR were retained for analysis. Cows were classified into three breed groups (BGs) based on their proportion of Holstein genetics: BG1 for cows with >93.7% Holstein genetics, BG2 for cows with 87.5–93.6% Holstein genetics, and BG for cows with <87.5% Holstein genetics. The age at calving in this population ranged from 18 to 50 months. The summary statistics of the analyzed traits are presented in Table 1.

### 2.2. Estimating the Genetic Parameters

The variance components for each trait were estimated by single-step genomic REML (ssGREML) with a multi-trait animal model. The AIREMLF90 and BLUPF90+ programs were used to estimate these components and solve the model equations [29]. The model included fixed effects, the animal’s additive genetic effects, and residual effects, as shown below:$$\left[\begin{array}{c} y_{1}\\ y_{2}\\ y_{3} \end{array}\right]=\left[\begin{array}{ccc} X_{1} & 0 & 0\\ 0 & X_{2} & 0\\ 0 & 0 & X_{3} \end{array}\right]\left[\begin{array}{c} b_{1}\\ b_{2}\\ b_{3} \end{array}\right]+\left[\begin{array}{ccc} Z_{1} & 0 & 0\\ 0 & Z_{2} & 0\\ 0 & 0 & Z_{3} \end{array}\right]\left[\begin{array}{c} u_{1}\\ u_{2}\\ u_{3} \end{array}\right]+\left[\begin{array}{c} e_{1}\\ e_{2}\\ e_{3} \end{array}\right]$$
where $y_{1}$, $y_{2}$, and $y_{3}$ represent the traits MY305, DO, and milk FPR; $b_{1}$, $b_{2},$ and $b_{3}$ represent the fixed effects of the breed group (3 levels), age at first calving (32 levels), and herd-month-year of calving (30,299 levels); $u_{1}$, $u_{2},$ and $u_{3}$ represent the random additive genetic effects vectors; and $e_{1}$**,**
$e_{2}$**,** and $e_{3}$ represent the random residual effects vectors. $X_{1},$ $X_{2},$ and $X_{3}$ and $Z_{1},$ $Z_{2},$ and $Z_{3}$ are the incidence matrices of fixed effects and additive genetic effects, respectively. The assumed covariance structures were as follows:$$Var\left[\begin{array}{c} u_{1}\\ u_{2}\\ u_{3} \end{array}\right]=A⊗\left[\begin{array}{ccc} \sigma_{u11}^{2} & \sigma_{u12} & \sigma_{u13}\\ \sigma_{u12} & \sigma_{u22}^{2} & \sigma_{u23}\\ \sigma_{u13} & \sigma_{e23} & \sigma_{u33}^{2} \end{array}\right],\;Var\left[\begin{array}{c} e_{1}\\ e_{2}\\ e_{3} \end{array}\right]=I⊗\left[\begin{array}{ccc} \sigma_{e11}^{2} & \sigma_{e12} & \sigma_{e13}\\ \sigma_{e12} & \sigma_{e22}^{2} & \sigma_{e23}\\ \sigma_{e13} & \sigma_{e23} & \sigma_{e33}^{2} \end{array}\right]$$
where matrix A denotes the additive relationship among the animals; matrix I denotes the identity; $\sigma_{u11}^{2}$, $\sigma_{u22}^{2}$, $\sigma_{u33}^{2}$, $\sigma_{u12}^{2}$, $\sigma_{u13}^{2}$, and $\sigma_{u23}^{2}$ denote the variances and covariances for additive genetic effects; and $\sigma_{e11}^{2}$, $\sigma_{e22}^{2}$, $\sigma_{e33}^{2}$, $\sigma_{e12}^{2}$, $\sigma_{e13}^{2}$, and $\sigma_{e23}^{2}$ denote the variances and covariances for residual effects.

The estimated heritability ($h_{i}^{2}$) and genetic correlations ($r_{gij}$) between MY305, DO, and milk FPR were calculated using the formula provided by Ravagnolo and Misztal [30], as follows:$$h_{i}^{2}=\frac{\sigma_{u_{ii}}^{2}}{\sigma_{u_{ii}}^{2}+\sigma_{e_{ii}}^{2}\;}\;\mathrm{and}\;r_{gij}=\frac{\sigma_{u_{ij}}}{\sqrt{\sigma_{u_{ii}}^{2}\times \sigma_{u_{jj}}^{2}}},$$
where $\sigma_{u_{ii}}^{2}$ represents the additive genetic variance for trait *i*, $\sigma_{e_{ii}}^{2}\;$ represents the residual variance, and $\sigma_{u_{ij}}\;$ defines the genetic covariance between traits *i* and *j*.

### 2.3. Estimating the Genomic Breeding Values

A weighted single-step genomic approach with multiple-trait genetic analysis was employed for genomic analysis. Instead of using the inverse of the numerator relationship matrix (A), the inverse of matrix (H) was applied in the mixed-model equation [27] described below.$$H^{-1}=A^{-1}+\left[\begin{array}{cc} 0 & 0\\ 0 & \tau G^{-1}-{\omega A}_{22}^{-1} \end{array}\right]$$

Here, $A^{-1}$ denotes the inverse of a pedigree-based relationship matrix for all the animals involved in the analysis; $G^{-1}$ denotes the inverse of a genomic relationship matrix (G); and $A_{22}^{-1}$ denotes the inverse of the pedigree-based relationship matrix specific to genotyped animals [31]. The weighting factors τ = 1.00 and ω = 0.50 were selected based on their ability to minimize bias during the preliminary validation study. Due to potential pedigree gaps, inbreeding was excluded from the A matrix, although its omission might have influenced the results. The genomic relationship matrix (G) and weights were constructed using the method described by VanRaden [31], as follows:$$G=\frac{{ZDZ}^{\prime }}{2\sum_{i=1}^{m}p_{i}(1-p_{i})\;}\mathrm{and}\;d_{i}=1.125^{\frac{\left|{\hat{s}}_{i}\right|}{S.D.({\hat{s}}_{i})}-2},$$

In this context, ***ZD**Z***′ represents the matrix of gene content adjusted for allele frequencies, D is a diagonal matrix containing weights for SNP variances, m denotes the total number of SNP markers, $p_{i}$ represents the frequency of the reference allele for the ith SNP, and $d_{i}$ represents the weight for the ith SNP.

The estimated breeding values (EBVs) and genomic breeding values (GEBVs) were calculated using the wssGBLUP option in the BLUPF90+ program. The accuracy of GEBVs was estimated from prediction error variance (PEV) functions available within the BLUPF90+ package.

### 2.4. Genome-Wide Association Study (GWAS)

The analyses were conducted using the wssGWAS approach. A similar model was applied in the wssGWAS to estimate the association between quantitative traits and individual single-nucleotide polymorphisms (SNPs). SNP effects were determined through an iterative process, identical to the method described by Wang et al. [32], and were implemented in POSTGSF90 Version 1.83 software [33]. The additive genomic breeding value ($\hat{u}$) was converted back to SNP effects ($\hat{s}$), considering their shared genomic variance $(\sigma_{u}^{2}$), using the following formula:$$\hat{s}={DZ}^{\prime }G^{-1}\hat{u}$$

The variance for each SNP was estimated as the proportion of variance explained by five adjacent SNPs relative to the additive genetic variance, using the following formula:$$\frac{Var({\hat{s}}_{j})}{{\hat{\sigma }}_{u}^{2}}\times 100\%=\frac{Var(\sum_{j=1}^{5}z_{j}{\hat{s}}_{j})}{{\hat{\sigma }}_{u}^{2}}$$

Here, ${Var(\hat{s}}_{j})\;$ represents the estimate of segment variance for the jth SNP, comprising 5 adjacent SNPs’ explained variance; ${\hat{\sigma }}_{u}^{2}$ denotes the total additive genetic variance; $z_{j}$ is the vector of gene content for the jth SNP across all individuals; and ${\hat{s}}_{j}$ is the estimated effect of the jth SNP within the *i*th region [34]. The threshold for significant SNP variance was determined by selecting SNPs with high explained variance that accounted for at least 5% of the total genomic variance. This threshold has been examined in cattle studies related to complex traits [35,36,37,38]. Manhattan plots for these windows were created using SAS Studio online v.3.81.

### 2.5. Identification of Candidate and Pleiotropic Genes

Potential candidate genes associated with MY305, DO, and milk FPR were identified by searching for genes near significant SNPs that exceeded the threshold. Genes located in the same genomic region across two or three traits were classified as pleiotropic genes. The distance between candidate genes and SNP locations was set to less than 50 kb (kilobase pairs), in accordance with the previous studies [39,40]. Gene identification was performed using the National Center for Biotechnology Information (NCBI) Map Viewer tool for the bovine genome, with UMD 3.1 assembly as the reference map (https://www.ncbi.nlm.nih.gov/gdv/browser/genome/?id=GCF_000003055.6 (accessed on 24 February 2025)) [35,41]. Further literature and database searches for all identified genes were conducted using the NCBI (https://www.ncbi.nlm.nih.gov/ (accessed on 24 February 2025)) and GeneCards (https://www.genecards.org/ (accessed on 24 February 2025)) platforms.

## 3. Results

### 3.1. Estimation of Heritability and Genetic Correlations

Table 2 presents the estimated variance components and heritability values for MY305, DO, and milk FPR. The additive genetic variances ($\sigma_{a}^{2}$) were 176,980 for MY305, 176.45 for DO, and 0.31 for milk FPR, reflecting the genetic contribution to the variation in these traits. The residual variances ($\sigma_{e}^{2}$) were 676,710 for MY305, 6099.85 for DO, and 2.99 for milk FPR, indicating the influence of environmental and non-genetic factors on their variation. The heritability estimates ($h^{2}$ ± SE) were 0.262 ± 0.005 for MY305, 0.029 ± 0.0004 for DO, and 0.102 ± 0.002 for milk FPR, indicating that a moderate proportion of variation in MY305 is due to genetic factors, whereas the genetic influence on DO and milk FPR is relatively low. The standard errors (SEs) for all three traits were lower than their respective heritability estimates, indicating statistically reliable estimates.

Table 3 presents the genetic $\left(r_{g}\right)$, phenotypic $\left(r_{p}\right)$, and environmental $\left(r_{e}\right)$ correlations among the MY305, DO, and milk FPR traits. A strong positive genetic correlation was observed between MY305 and DO (0.559). In contrast, strong negative genetic correlations were identified between MY305 and milk FPR (−0.306) and between DO and milk FPR (−0.501), indicating an antagonistic genetic relationship among these traits. Phenotypic and environmental correlations among the three traits were generally low, with values below 0.1 indicating that shared environmental and phenotypic influences have a minimal impact on the relationships between MY305, DO, and milk FPR.

### 3.2. Estimation of GEBVs Using ssGBLUP and wssGBLUP

Table 4 presents the average genomic estimated breeding values (GEBVs) obtained using ssGBLUP and wssGBLUP for MY305, DO, and milk FPR, categorized by dataset and breed group. wssGBLUP yielded higher GEBV estimates than ssGBLUP for all three traits. Generally, the GEBVs for MY305 and DO decreased as the percentage of Holstein genetics decreased, whereas the milk FPR increased in both the ssGBLUP and wssGBLUP analyses across all animal datasets. For MY305, both methods produced negative GEBVs, but wssGBLUP (−8.983 to −8.259) resulted in less negative values than ssGBLUP (−12.492 to −11.792), suggesting a potential positive impact on milk yield. A similar trend was observed in the bull dataset, where wssGBLUP produced fewer negative GEBVs for MY305 and better GEBVs for DO and milk FPR. Among dams, wssGBLUP generally improved GEBVs across all traits, although differences in DO and milk FPR were less pronounced than those observed for MY305.

Figure 1 presents the averaged accuracy of GEBVs from the two genomic prediction methods, ssGBLUP and wssGBLUP, across all animal groups: all animals, bulls, and dams. wssGBLUP consistently outperformed ssGBLUP, demonstrating higher accuracy across all traits and dataset groups. For instance, in the all-animals group, the accuracy for MY305 increased from 0.138 (ssGBLUP) to 0.176 (wssGBLUP). Similarly, the accuracy for DO improved from 0.112 to 0.124, and for milk FPR, it improved from 0.094 to 0.111. These improvements remained stable across all groups, indicating the robust and reliable advantage of wssGBLUP over ssGBLUP. Across the breed groups, bulls and dams exhibited variation in trait performance under both the ssGBLUP and wssGBLUP models. Bulls generally demonstrated higher genetic potential for MY305, while dams showed slightly greater variability in DO and milk FPR.

### 3.3. Weighted Single-Step Genome-Wide Association Study

The wssGWAS analysis was successfully conducted using genotypic data from 43,284 SNPs obtained from 868 cows for the MY305, DO, and milk FPR traits. The results are illustrated in the Manhattan plots in Figure 2 and Figure 3, where each color represents a specific chromosome. This study identified 14 SNPs associated with MY305 across seven chromosomes, 19 SNPs linked to DO across five chromosomes, and 16 SNPs associated with milk FPR across five chromosomes, adhering to a distance constraint of less than 50 kb (Table 5, Table 6 and Table 7, respectively).

### 3.4. Identification of Pleiotropic Genes for MY305, DO, and Milk FPR in Thai-Holstein Crossbred Cattle

Following the candidate gene analysis, we identified genes located in the same genomic region across all three traits, suggesting potential pleiotropic effects (Figure 4). Fourteen genes were identified within the same region for MY305, DO, and milk FPR. Additionally, 17 genes were shared between MY305 and DO, whereas 14 genes were common between DO and milk FPR. Furthermore, we conducted a relationship analysis between SNP variance and SNP effects for MY305, DO, and milk FPR (Figure 5, Figure 6 and Figure 7). A positive linear relationship was observed between the SNP variance in MY305 and DO. In contrast, a negative linear relationship was found between the SNP variance in DO and milk FPR, as well as between that in MY305 and milk FPR. Similar patterns were observed in SNP effect relationships among the three traits.

## 4. Discussion 

### 4.1. Heritabilities

Heritability estimates (*h*^2^) are crucial for understanding the genetic influence on these traits and their potential for improvement through selective breeding. The heritability of MY305 (0.262 ± 0.005) suggests that a moderate proportion of variation in this trait is attributable to genetic factors. While genetic selection can enhance MY305, effective management strategies are necessary, especially in challenging environments, to achieve substantial progress (Table 2). The high additive genetic variance ($\sigma_{u}^{2}$) further supports the potential for genetic improvement in MY305. The heritability estimate for MY305 observed in this study aligns with values previously reported in the literature, ranging from 0.19 to 0.27 for Holstein cows [38,39,40,41] and from 0.24 to 0.27 for Guzerat cows [42,43]. Beyond breed differences, the lactation stage also influences heritability variations. Pangmao et al. [44] reported that the heritability for MY305 was generally higher during the first lactation compared to subsequent lactations. Additionally, differences in management practices, climate conditions, and methodological approaches across populations can significantly affect MY305 estimates and heritability [45].

However, the low heritability of DO (0.029 ± 0.0004) suggests that most of the variation in this trait within the population was due to management practices, with genetic factors playing a minor role. As a result, selective breeding may have a limited impact on improving DO, and alternative strategies, such as enhancing environmental conditions and reproductive management, may be more effective. The heritability of DO observed in this study was lower than the values reported for Holstein cows, which range from 0.04 to 0.07 [46,47,48,49], and for Russian Black-and-White cattle (0.07) [50]. Additionally, it was lower than the values previously reported for the same population in 2015 (0.04 to 0.05) [19], but comparable to values reported in a more recent study [22]. The low heritability value may be due to the strong influence of fixed factors on DO and the high residual variance observed in this study ($\sigma_{e}^{2}$). A previous study identified factors affecting DO, including Holstein genetic proportions and heat stress in hot climates [51,52]. Studies from other subtropical regions have suggested that environmental stress can increase genetic variability in dairy cattle [49,53]. In Thailand’s tropical climate, heat stress poses a significant challenge for livestock, potentially compromising reproductive performance. Additionally, variations in the month and year of calving across different populations may have contributed to the low heritability estimates of DO. To improve this trait, strategies such as better reproductive management, heat stress control, skilled artificial insemination, modern milking facilities, and optimized nutrition should be implemented [54,55].

Low heritability (0.102 ± 0.002) was observed for milk FPR, indicating that fixed factors and residual variance were the primary contributors to its variation within the population. Similar heritability values have been reported in Thai-Holstein crossbreeds (0.17) [17] and Nordic cattle (0.14) [19]. In contrast, higher heritability values have been observed in Holstein cows, ranging from 0.25 to 0.54 [11,56,57,58,59]. The variation in milk FPR heritability estimates across studies may be due to differences in breed, estimation methods, and model effects. Genetic differences between cows and environmental factors such as heat stress have also been reported to influence milk FPR [56]. Additionally, a study utilizing random regression test-day models reported higher milk FPR heritability values [59]. Genetic improvement of milk FPR through selection requires careful management strategies to achieve meaningful progress, particularly in preventing metabolic disorders, such as acidosis and ketosis.

### 4.2. Genetic Correlations Between MY305, DO, and Milk FPR

Table 3 shows the positive genetic correlation between MY305 and DO (0.559), suggesting that cows with a higher milk yield over 305 days tended to have extended DO. Consequently, selection for increased milk production may compromise fertility. This finding aligns with previous research on Holstein cows [60,61,62], showing that nutrient allocation prioritizes milk production over reproductive functions [63]. Additionally, management strategies that delay the first insemination in high-yielding cows may further contribute to prolonged DO [64]. Moreover, high-producing dairy cows often experience a postpartum negative energy balance (NEB), which can significantly impair fertility by delaying ovulation [13,65]. These results emphasize the need for balanced breeding strategies to improve milk production while maintaining reproductive efficiency.

A negative genetic correlation (−0.306) was identified between MY305 and milk FPR, indicating that cows with higher MY305 often have lower milk FPR. Similar genetic correlations between MY305 and milk FPR have been reported in Thai-Holstein crossbreeds [17,56] and Nordic Red cattle [19]. Negussie et al. [19] found that genetic correlations between test-day FPR and MY during early lactation at 15, 30, and 60 days in milk (DIM) were small but positive (0.01 to 0.13), and after 60 DIM, these correlations became predominantly negative (0.01 to −0.22). These positive genetic correlations in early lactation suggest that high-producing cows mobilize body reserves to meet the energy demands of peak milk production, leading to a higher FPR. However, as lactation progresses, the genetic correlation between FPR and MY declines or becomes negative, indicating recovery from NEB. This trend is consistent with the findings in Canadian Holsteins [18] and German Holsteins [66]. In Thai dairy cattle, this condition may arise because of less intensive management during late lactation compared to early lactation, particularly among dairy farmers who continue to use traditional practices. As a result, these animals often experience inadequate nutrition management [67].

One notable finding from this study is the negative genetic correlation between DO and milk FPR, which indicates that cows with higher DO tend to have a lower milk FPR. These results contradict previous studies that reported low-to-moderate positive genetic correlations between DO and milk FPR [17,19]. This may be attributed to a lack of awareness regarding the nutritional needs of dairy cows during lactation, particularly during the late lactation phase. An extremely low milk FPR indicates acidosis, a condition often caused by excessive concentrate feeding without sufficient high-quality forage. Overfeeding with concentrates leads to excessive starch fermentation, which increases lactic acid production and lowers rumen pH. This can negatively impact reproductive performance, resulting in infertility, prolonged calving-to-conception intervals, and increased incidence of lameness [12,15,68]. Alternatively, the negative genetic correlation between DO and milk FPR may be influenced by differences in the time from calving to first service between trained officers and traditional farmers, which affects DO variation among cows [67].

In our study, the phenotypic and environmental correlations among MY305, DO, and milk FPR were very low, with both positive and negative values close to zero, indicating that these traits share minimal common influences at the observable and environmental levels. This suggests that changes in MY305 are unlikely to have a direct impact on DO and milk FPR, and shared environmental factors do not significantly affect these traits together. As a result, improvements in milk yield management or genetic selection may not necessarily affect fertility and animal health.

### 4.3. GEBVs for MY305, DO, and Milk FPR

The breed group (BG) effects indicated that a higher proportion of Holstein genetics was associated with increased MY305, longer DO, and lower milk FPR across both methods (Table 4). Holsteins have superior genetic potential for milk production compared to many other breeds, which explains why animals in BG1 exhibited the highest MY305. Studies have consistently shown that Holsteins outperform other breeds in terms of milk production [69,70,71]. However, animals with higher Holstein genetics also tend to have a longer DO, possibly because of their reduced adaptability to tropical environments. This finding aligns with the study by Pongpiachan et al. [69], who reported that purebred Holsteins exhibited lower reproductive efficiency, even when specialized management strategies were employed to mitigate the effects of tropical climates and enhance their diet. Consequently, crossbreeding has been proposed as a strategy to counteract reproductive decline associated with “Holsteinization” [72]. A previous study found that crossbred cows exhibited improved reproductive performance, including shorter DO and calving intervals (CIs), higher conception rates at 28 days, and reduced incidences of mastitis, ketosis, and endometritis [73]. Additionally, high-production Holsteins often experience greater energy deficits, leading to increased mobilization of body fat. This metabolic shift can alter the fat-to-protein ratio (FPR) in milk, often resulting in a lower FPR [67].

The GEBV results presented in Table 4 demonstrate that wssGBLUP consistently produced higher GEBVs than ssGBLUP, particularly for MY305 and DO, indicating that wssGBLUP captured genetic effects in the dataset more effectively. In the all-animal dataset, both methods yielded negative GEBVs for MY305, but the wssGBLUP values (−8.983 to −8.259) were closer to zero, suggesting enhanced GEBV accuracy. A similar trend of improvement was observed in the bull dataset, whereas in the dam dataset, the improvements were less pronounced for DO and milk FPR than for MY305. The GEBV for MY305 in this population was generally negative, in contrast to that of Russian Black-and-White cattle, which was reported to be 0.88 [50]. A negative average GEBV for MY305 suggests that the population’s genetic potential for milk yield was below the baseline or the genetic mean defined by the reference population. This may have resulted from factors such as population structure or the genetic merit of the reference group used for comparison. Research has shown that the composition of the reference population plays a crucial role, as it directly affects the estimated breed composition and subsequent GEBVs [74]. In this study, the reference population consisted of high-performing breeds, which might have led to lower GEBVs for animals from breeds with a lower genetic potential for MY305.

The positive GEBVs for DO indicate that animals have a higher genetic potential for longer reproductive intervals, which may not be desirable based on breeding goals favoring shorter calving intervals. However, the mean GEBVs for DO in the all-animal dataset were lower than those reported in Russian Black-and-White cattle (3.25 to 4.14) [50] but comparable to recent findings of 0.266 to 0.274 [22]. This suggests that the Thai-Holstein population has better genetic values for the DO trait than the Russian Black-and-White cattle population. This improvement may be attributed to the genetic advantages of crossbred cattle, which exhibit superior reproductive efficiency and overall performance compared with purebred Holsteins. For milk FPR, the GEBVs remained consistently negative or near zero across both the ssGBLUP and wssGBLUP methods, indicating that both models provided similar GEBV accuracy for this trait.

The wssGBLUP method consistently demonstrated a higher accuracy than ssGBLUP across all traits and groups (Figure 1). For MY305, the accuracy increased by 27.54%, improving from 0.138 with ssGBLUP to approximately 0.176–0.177 with wssGBLUP. This highlights both the higher heritability of MY305 and the enhanced ability of wssGBLUP to capture key genetic markers compared to other traits. In contrast, DO and milk FPR exhibited smaller accuracy gains, with the averaged GEBV accuracy increasing from 0.112–0.113 to 0.124–0.125 (10.71%), and from 0.094 to approximately 0.110–0.112 (17.02%), respectively. These findings suggest that wssGBLUP improved accuracy by emphasizing key genetic regions, with notable gains for highly heritable traits, such as MY305. These results align with the findings of previous studies, where wssGBLUP provided greater accuracy for production traits, while yielding smaller gains for traits with lower genetic influence. Zhang et al. [75] reported that wssGBLUP enhances accuracy by assigning different weights to SNPs, making it particularly effective for highly heritable traits in dairy cattle. The wssGBLUP method has been shown to effectively identify SNPs linked to traits such as protein content and to enhance the accuracy of GEBVs in Canadian Holstein cattle [76]. Additionally, a study on Hanwoo beef cattle found that wssGBLUP improved the prediction accuracy for carcass traits such as carcass weight, eye muscle area, and yearling weight [77]. In pig breeding, a study revealed that wssGBLUP offered improved estimation reliability compared to ssGBLUP for meat, fattening, and reproductive traits [78].

wssGBLUP is superior to ssGBLUP because it can assign weights to single-nucleotide polymorphisms (SNPs), optimize predictions for specific traits, and iteratively refine accuracy. According to Teissier et al. [79], wssGBLUP assigns different weights to SNPs based on their estimated effects, enabling the more precise identification of quantitative trait loci (QTLs) with significant impacts on traits. By emphasizing SNPs associated with major genes, wssGBLUP enhanced the accuracy of GEBVs for traits strongly influenced by these genes. Additionally, the iterative weighting process of wssGBLUP allows it to adapt to the genetic architecture of different traits, making it highly flexible in modeling both polygenic traits and those controlled by a few major genes [77].

### 4.4. Identification of Genomic Regions and Candidate Genes

Manhattan plots of the SNP effects from GEBVs for MY305, DO, and milk FPR are shown in Figure 2. The peaks of the Manhattan plots highlight impactful SNPs, with positive values at the top and negative values at the bottom. Figure 3 presents Manhattan plots illustrating the percentage of additive genetic variance accounted for by the five SNP moving windows. Using a threshold of at least 5% of the total genetic variance, 14 SNPs were significantly associated with MY305 across seven chromosomes (BTA 4, 6, 13, 18, 24, and 25), 19 SNPs with DO across five chromosomes (BTA 2, 5, 13, 18, and 25), and 16 SNPs with milk FPR across five chromosomes (BTA 5, 18, 19, 21, and 25), all within a distance constraint of less than 50 kb (Table 6 and Table 7).

After comparing the results with NCBI databases, we identified 24 candidate genes associated with MY305, 46 genes associated with DO, and 33 genes associated with milk FPR. However, four genes from MY305 (*LOC618297*, *LOC101906304*, *LOC529511*, and *LOC786948*), nine genes from DO (*LOC101905166*, *LOC107133069*, *LOC101902869*, *LOC100138951*, *LOC618463*, *LOC100141212*, *LOC101905312*, *C25H16orf58*, and *LOC783313*), and six genes from milk FPR (*LOC101905166*, *LOC100138951*, *LOC618463*, *LOC100141212*, *LOC100196898*, and *LOC786948*) have not been characterized (Table 6 and Table 7). We also identified seven genes from MY305 (*LOC618297*, *LOC101906304*, *PPARGC1A*, *VSIG10L*, *PARD6G*, and *LMF1*), ten genes from DO (*TNS1*, *SOX10*, *LOC101905166*, *BAIAP2L2*, *PDYN*, *LOC618463*, *VSIG10L*, *LMF1*, *ITGAD*, and *SEPT14*), and nine genes from milk FPR (*SOX10*, *LOC101905166*, *BAIAP2L2*, *LOC618463*, *VSIG10L*, *LOC100196898*, *MGAT5B*, *MGAT5B*, and *LINGO1*) at the target SNP location. Hajihosseinlo et al. [80] found that the r^2^ value tends to decrease as the distance between SNP pairs increases, implying that SNPs positioned within 1 Mb are more likely to exhibit strong and consistent associations with QTLs.

Candidate gene analysis revealed genes in the same genomic region across all three traits, suggesting potential pleiotropic effects (Figure 4). Specifically, 14 genes were shared among MY305, DO, and milk FPR; 17 genes were shared between MY305 and DO; 14 genes were shared between MY305 and milk FPR; and 26 genes were shared between DO and milk FPR. According to Gratten and Visscher [81], pleiotropy is a genetic phenomenon whereby a single DNA variant influences multiple traits. This indicates that when selection targets one trait, other traits often change over generations. This response is driven by genetic correlations that reflect the combined genome-wide effects of pleiotropy at shared genetic loci. Identifying pleiotropic genes associated with MY305, DO, and milk FPR is challenging, owing to the complexity of these traits. Although limited research has directly connected these traits to pleiotropic genes, some studies have examined genes that affect multiple production traits [35,82] and multiple fertility traits in dairy cattle [83,84].

### 4.5. Pleiotropy and Candidate Genes for MY305, DO, and Milk FPR

Pleiotropy occurs when one gene affects multiple phenotypic traits. This study identified genes influencing MY305, DO, and milk FPR simultaneously. The pleiotropic effects of these genes are shown through their SNP variance and effect relationships (Figure 5, Figure 6 and Figure 7).

Figure 5A, Figure 6A, and Figure 7A show positive relationships between SNP variance for MY305 and DO, FPR and DO, and MY305 and FPR, respectively. Most SNP variances are clustered near the origin, indicating that most SNPs have low variances across all traits, while a few SNPs exhibit strong pleiotropic genetic influences. Influential SNPs can be found in all figures, indicating significant pleiotropic SNPs for all three relationships. It can be observed that SNPs affecting high MY305 variation also affect high DO variation (Figure 5A). SNPs influencing high FPR variation affect moderate DO variation (Figure 6A). However, SNPs influencing high DO and FPR variation might contribute independently (Figure 7A).

Figure 5B, Figure 6B, and Figure 7B demonstrate strong relationships between SNP effects for MY305 and DO, FPR and DO, and MY305 and FPR, respectively. Positive collinearity can be observed in SNPs affecting MY305 and DO, indicating that SNPs with large effects on MY305 also extend to DO (Figure 5B). A negative relationship can be found in SNPs affecting FPR and DO, showing that SNPs with large negative effects on FPR tend to increase DO (Figure 6B). Similarly, SNPs influencing MY305 and FPR have strong negative correlations, suggesting that high MY305 effects decrease the FPR (Figure 7B). This implies that selecting cows for high milk yield may lower the FPR and prolong DO, posing a challenge, as a greater value for DO is linked to extended calving intervals and reduced fertility. These finding confirm previous studies reporting antagonistic genetic correlations between milk yield and fertility [85,86], indicating that higher milk yield might reduce the fat-to-protein ratio, affecting milk composition and metabolic efficiency.

These findings highlight the need for multi-trait selection strategies to balance milk yield, fertility, and metabolic efficiency. Breeding programs should use optimal selection indices to improve MY305, DO, and milk FPR. High-variance SNPs in these traits suggest strong genetic influences, making them key targets for marker-assisted selection (MAS). Identifying beneficial SNPs for all three traits can refine breeding strategies. Additionally, pleiotropic genes associated with all three traits highlight the genetic connections among production, fertility, and health traits.

The results reveal fourteen genes within the same region, indicating pleiotropy among MY305, DO, and milk FPR (*SIGLECL1*, *IGLON5*, *VSIG10L*, *ETFB*, *NKG7*, *CLDND2*, *LIM2*, *SOX8*, *SSTR5*, *TEKT4*, *C1QTNF8*, *CACNA1H*, *LOC786948*, and *TPSB2*). Some genes have been reported to be pleiotropic and influence production, reproduction, and health traits across various species. Recent studies highlight *SIGLECL1*’s role in animal reproduction and immune regulation, with its expression detected in the male reproductive tracts of mice, rats, and bovine sperm [87,88]. Additionally, SIGLEC family polymorphisms have been associated with milk yield, DO, and calving intervals in cattle [89,90].

The following genes play a role in reproduction in some species. *LIM2* plays a role in spermatogenesis, with studies in mice showing that *Limk2*-deficient individuals exhibit impaired testicular development and increased germ cell apoptosis [91]. This gene also protects spermatogenic cells from stress-induced damage [91]. *TEKT4* is expressed in male germ cells, and is essential for sperm motility in mice [92,93]. Additionally, *TEKT4* has been identified as a key gene associated with sperm motility in Brahman cattle, based on proteomic studies [94].

Three genes, *SSTR5*, *ETFB*, and *SOX8*, have been linked to animal production and growth traits. SSTR5 polymorphisms in sheep are associated with growth traits, making them potential molecular markers for selective breeding [95]. At the same time, *ETFB* has been identified as a key gene influencing meat quality in Qinchuan cattle [96]. Copy number variations (CNVs) of *SOX8* genes in yaks are significantly associated with growth traits such as withers height and chest girth [97].

Among the genes related to immunity and behavior, *IGLON5* plays a critical role in immune function in cattle [98,99]. *NKG7* enhances cytotoxic activity in CD8+ T cells and NK cells, promoting T cell accumulation [100,101]. *CLDND2* plays a role in immune responses in cattle [102]. *CACNA1H* knockout in mice leads to autistic-like behaviors [103]. *TPSB2* is linked to immune cells in adipose tissue in Holstein–Friesian cows [104]. Additionally, three genes (*PDYN*, *SIRPA*, and *LMF1*) were found to be shared between MY305 and DO. *PDYN* shows signs of positive selection in dairy cattle, indicating its role in reproductive traits [105]. *SIRPA* is a marker of spermatogonial stem cells, embryogenesis, and gametogenesis in mice [106,107,108]. *LMF1* is essential for the post-translational activation of lipoprotein lipase and other enzymes [109,110].

The results indicate that DO and milk FPR share 17 common genes (*MICALL1*, *C5H22orf23*, *SOX10*, *POLR2F*, *LOC101905166*, *PICK1*, *SLC16A8*, *BAIAP2L2*, *PLA2G6*, *LOC100138951*, *LOC618463*, and *LOC100141212*); most genes influence health and immunity, while others impact fertility. *BAIAP2L2* genes are essential for mechanotransduction [111]. *PLA2G6* encodes *iPLA2β*, related to immunity and membrane homeostasis [112]. *LOC618463* is associated with livability in Holstein cattle [113], and *LOC100138951* is linked to calf survival in Nordic Holstein cattle [114]. The *SLC16A8* gene family includes 14 monocarboxylate transporters vital for metabolic processes [115]. *LOC618463* is also a target gene for calving traits [116]. *SOX10* is involved in sex determination [117], and *PICK1* is essential for male fertility [118].

Candidate genes in MY305 include *LOC618297*, *SHFM1*, *LOC101906304*, *PPARGC1A*, *LOC529511*, *PARD6G*, and *ADNP2. PPARGC1A* polymorphisms are linked to milk fat yield [119] and birth weight in Holsteins [120], with specific SNPs affecting milk traits in Iranian Holsteins [121]. *PARD6* is associated with the Hippo signaling pathway [122,123], and *ADNP2* is highly expressed in embryonic brain tissues [124].

Seventeen candidate genes influencing DO were identified, with key roles in reproduction (*TNS1*, *MAFF*, *LOC107133069*, *LOC101902869*, *ITGAD*, *COX6A2*, *ARMC5*, *LOC101905312*, *TGFB1I1*, *SLC5A2*, *RUSF1*, *AHSP*, *LOC783313*, *OR7D4*, *SEPT14*, *ZNF713*, and *MRPS17*). *ITGAD* is linked to fertility and tick resistance [124,125,126], and *TGFB1I1* is important for ovarian development [127]. Genes from the TGF-β pathway are essential for embryogenesis [128]. The SLC5 family, especially *SLC5A1* and *SLC5A2*, affects glucose transport and litter size in sheep [129]. *OR7D4* in horses and stallion testes plays a role in reproductive functions [130]. *MRPs* like *MRPS17* are vital for embryogenesis [131], and *MRPS17* and *SEPTIN14* are candidates for fertility traits in cattle [83]. *C25H16orf58* and *SLC45A2* are associated with clinical ketosis in Holsteins and heat tolerance in Chinese cattle, respectively [132,133].

Finally, five genes (*LOC100196898*, *MGAT5B*, *MFSD11*, *LINGO1*, and *ODF3L10*) were associated with milk FPR. In dairy cattle, milk FPR is linked to NEB and the overall health condition. *MGAT5* is crucial for the synthesis of complex N-glycans and is essential for various biological processes. *MGAT5*-deficient mice exhibit complex phenotypes, including susceptibility to autoimmune diseases and reduced cancer progression [134,135]. Additionally, *MGAT5* has been associated with mastitis resistance [136]. *MFSD11* is widely expressed in the brain and periphery, particularly in the neurons, and its expression is altered by changes in energy balance in mice [137]. *LINGO1*, a protein selectively expressed in the central nervous system, functions as a negative regulator of oligodendrocyte differentiation, myelination, neuronal survival, and axonal regeneration and plays a key role in neural health [138]. Research on agonistic behavior in Lidia cattle has also identified genomic regions containing *LINGO2* that are linked to behavioral traits [139]. At the same time, *ODF3L10* (also known as Odf3) is primarily associated with sperm tail outer dense fibers (ODFs) in animals with internal fertilization [140]. Most of these genes contribute to health-related traits in cattle, including neurological function, energy balance, immune response, and structural integrity.

## 5. Conclusions

In conclusion, this study highlights the potential of wssGBLUP to improve GEBV accuracy in Thai-Holstein crossbred dairy cattle. The identification of pleiotropic genes provides valuable insights into the genetic relationships between milk yield, fertility, and health traits, enabling more informed breeding decisions. Heritability estimates suggest a moderate genetic influence on milk yield, whereas fertility and milk FPR exhibit lower heritability, emphasizing the challenges of genetic improvement in these traits. Furthermore, genetic correlations reveal trade-offs in selection, particularly a negative relationship between milk yield and fertility, which must be carefully managed in breeding strategies. wssGBLUP outperformed ssGBLUP, making it a promising alternative for multi-trait genomic prediction. Breeding strategies should integrate genomic markers associated with desirable pleiotropic effects, enabling balanced selection for high milk yields without compromising fertility and health traits. To ensure sustainable genetic progress, further functional validation of the candidate genes is essential. Incorporating these validated markers into selection programs can optimize breeding decisions, improve reproductive efficiency, and enhance the overall productivity of tropical dairy cattle. Ultimately, this approach could contribute to long-term genetic gains and economic sustainability in the dairy industry.

## Figures and Tables

**Figure 1 animals-15-01320-f001:**
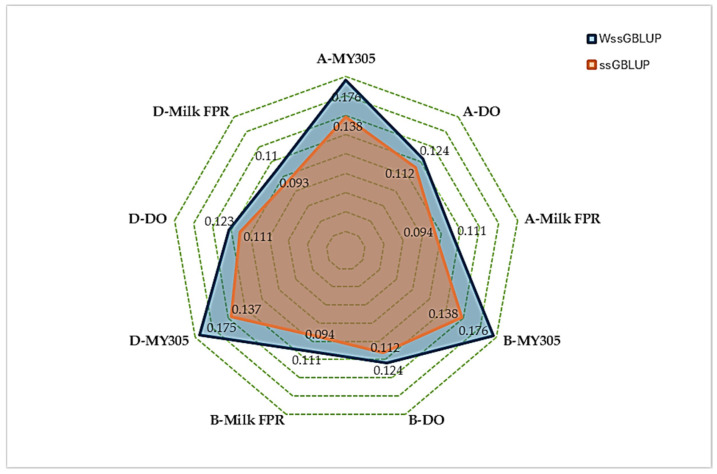
Comparison of GEBV accuracy between ssGBLUP and wssGBLUP in all animal (A), bull (B), and dam (D) datasets.

**Figure 2 animals-15-01320-f002:**
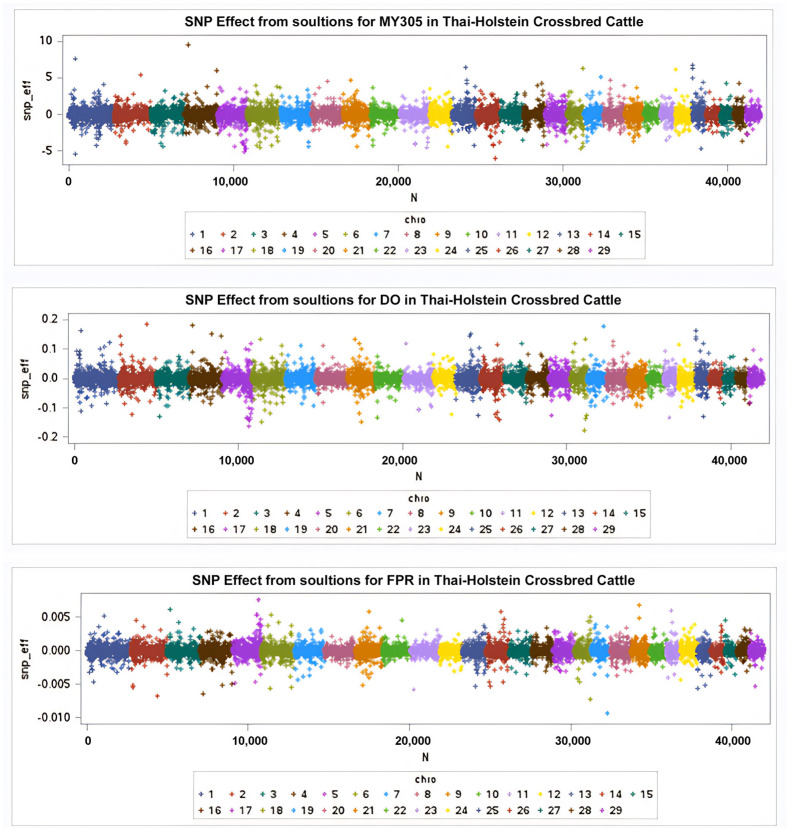
SNP effects derived from GEBVs for MY305, DO, and milk FPR, respectively.

**Figure 3 animals-15-01320-f003:**
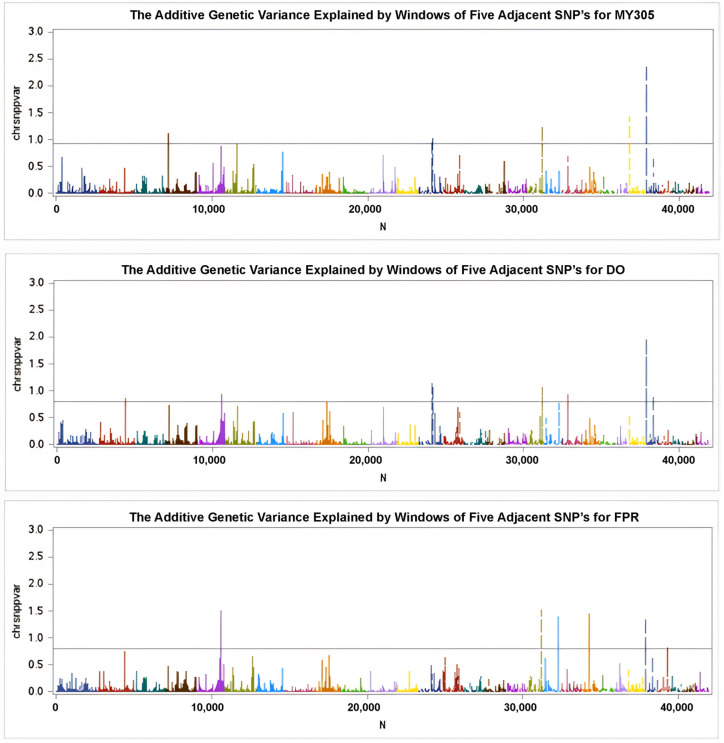
Manhattan plots of the additive genetic variance explained by windows of five adjacent SNPs for MY305, DO, and milk FPR, respectively. The *y*-axis shows the SNP variance across the genome, while the *x*-axis indicates SNPs’ positions on individual chromosomes. The horizontal line marks the threshold level, suggesting significance at a minimum of 5% of the total genetic variance.

**Figure 4 animals-15-01320-f004:**
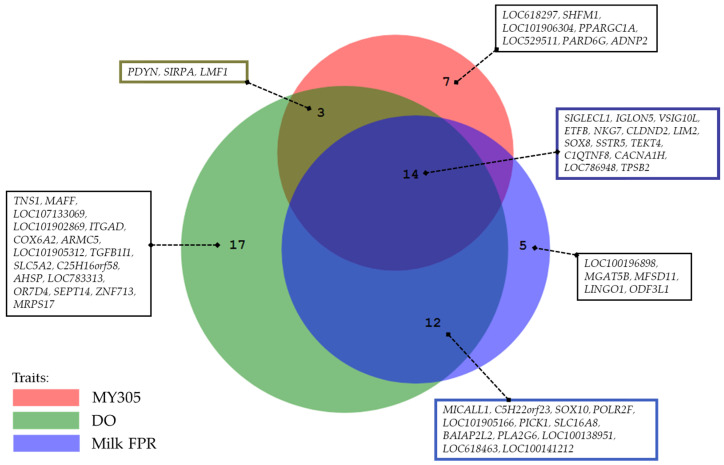
Pleiotropic genes identified for MY305, DO, and milk FPR.

**Figure 5 animals-15-01320-f005:**
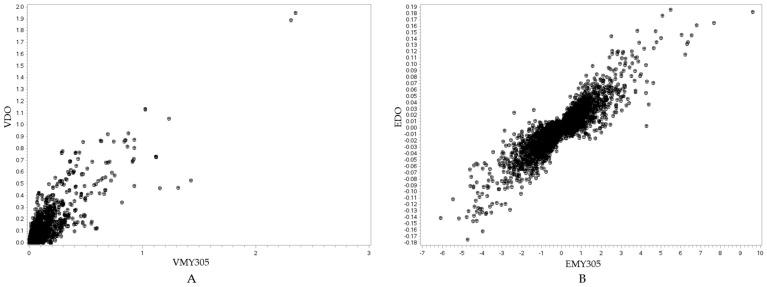
Visualization of the relationship between the SNP variance of the MY305 and DO traits (*r* = 0.8728, *p* < 0.0001) (**A**), and the SNP effects of the MY305 and DO traits (*r* = 0.8915, *p* < 0.0001) (**B**). In (**A**), the *y*-axis represents the SNP variance for the DO trait (VDO), while the *x*-axis represents the SNP variance for the MY305 trait (VMY305). In (**B**), the *y*-axis represents the SNP effect for the DO trait (EDO), while the *x*-axis represents the SNP effect for the MY305 trait (EMY305).

**Figure 6 animals-15-01320-f006:**
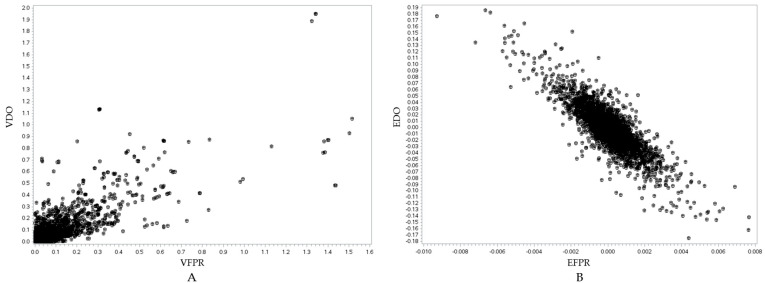
Visualization of the relationship between the SNP variance of the DO and milk FPR traits (*r* = 0.8219, *p* < 0.0001) (**A**) and the SNP effects of the DO and milk FPR traits (*r* = −0.8394, *p* < 0.0001) (**B**). In (**A**), the *y*-axis represents the SNP variance for the DO trait (VDO), while the *x*-axis represents the SNP variance for the milk FPR trait (VFPR). In (**B**), the *y*-axis represents the SNP effect for the DO trait (EDO), while the *x*-axis represents the SNP effect for the milk FPR trait (EFPR).

**Figure 7 animals-15-01320-f007:**
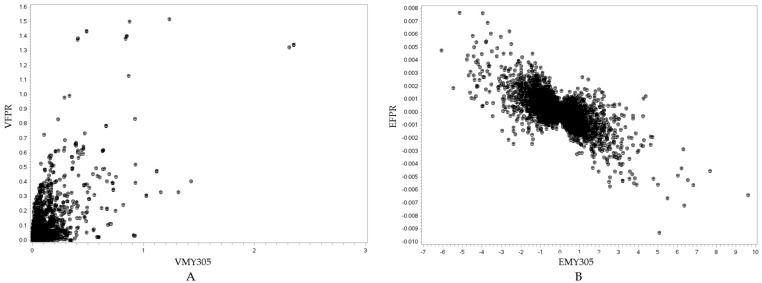
Visualization of the relationship between the SNP variance of the milk FPR and MY305 traits (*r* = 0.6993, *p* < 0.0001) (**A**), and the SNP effects of the milk FPR and MY305 traits (*r* = −0.7018, *p* < 0.0001) (**B**). In (**A**), the *y*-axis represents the SNP variance for the milk FPR trait (VFPR), while the *x*-axis represents the SNP variance for the MY305 trait (VMY305). In (**B**), the *y*-axis represents the SNP effect for the milk FPR trait (EFPR), while the *x*-axis represents the SNP effect for the MY305 trait (EMY305).

**Table 1 animals-15-01320-t001:** Data structure of MY305, DO, and milk FPR for Thai-Holstein crossbred cattle.

Categories	Number	Mean	SD	Minimum	Maximum
305-day milk yield, kg	18,843	3529.22	1144.85	514	9089
BG1	5672	3640.78	1164.99	567	8806
BG2	8004	3578.22	1118.23	514	8655
BG3	5167	3330.86	1138.74	692	9089
Days open, days	48,274	156.29	81.92	20	400
BG1	12,742	159.56	83.46	20	400
BG2	22,723	156.15	81.47	20	400
BG3	12,809	153.28	81.05	20	400
Fat-to-protein ratio	18,843	1.13	0.28	0.31	3.05
BG1	5672	1.12	0.28	0.31	3.05
BG2	8004	1.13	0.28	0.33	2.91
BG3	5167	1.13	0.27	0.32	2.62
Animals with pedigrees, n	87,958	-	-	-	-
Herd-month-year of calving, n	30,299	-	-	-	-

SD, standard deviation; BG, breeding group (BG1 = greater than 93.7% HF genetics; BG2 = 87.5 to 93.6% HF genetics; and BG3 = less than 87.5% HF genetics).

**Table 2 animals-15-01320-t002:** Variance components and heritability values for MY305, DO, and milk FPR.

Variance Components and Heritability Values	MY305	DO	Milk FPR
$\sigma_{u}^{2}$	176,980	176.45	0.31
$\sigma_{e}^{2}$	499,730	5923.40	2.68
$\sigma_{t}^{2}$	676,710	6099.85	2.99
$h^{2}$ ± SE	0.262 ± 0.005	0.029 ± 0.0004	0.102 ± 0.002

$\sigma_{u}^{2}$ = additive genetic variance; $\sigma_{e}^{2}$ = residual variance; $h^{2}$ = heritability; SE = standard error.

**Table 3 animals-15-01320-t003:** Genetic, phenotypic, and environmental correlations between MY305, DO, and milk FPR.

Traits	$r_{g}$ ± SE	$r_{p}$ ± SE	$r_{e}$ ± SE
MY305—DO	0.559 ± 0.006	0.002 ± 0.007	−0.06 ± 0.007
MY305—Milk FPR	−0.306 ± 0.007	−0.048 ± 0.007	0.003 ± 0.007
DO—Milk FPR	−0.501 ± 0.006	0.002 ± 0.007	0.03 ± 0.007

$r_{g}$ = genetic correlation; $r_{p}$ = phenotypic correlation; $r_{e}$ = environmental correlation; SE = standard error.

**Table 4 animals-15-01320-t004:** Comparison of breed group effects and breed group averages of EBVs and GEBVs for MY305, DO, and milk using ssGBLUP and wssGBLUP.

Dataset	Breed Group	ssGBLUP			wssGBLUP		
		MY305	DO	Milk FPR	MY305	DO	Milk FPR
BG effects	1	94.418	8.113	−0.007	1227.296	56.038	3.799
	2	38.265	4.082	0.051	1170.026	52.033	3.859
	3	0.000	0.000	0.000	1128.989	47.848	3.809
All animals	1	−12.492	0.237	−0.011	−8.983	0.279	−0.012
	2	−11.894	0.225	−0.011	−8.716	0.262	−0.012
	3	−11.792	0.225	−0.013	−8.259	0.271	−0.014
Bulls	1	−12.144	0.226	−0.012	−8.123	0.322	−0.014
	2	−10.305	0.262	−0.013	−7.913	0.277	−0.012
	3	−10.655	0.258	−0.014	−7.285	0.276	−0.014
Dams	1	−8.714	0.315	−0.015	−7.976	0.327	−0.014
	2	−8.401	0.317	−0.015	−8.101	0.270	−0.012
	3	−10.867	0.245	−0.013	−6.721	0.300	−0.015

BG = breed group (BG1 = greater than 93.7% HF genetics; BG2 = 87.5 to 93.6% HF genetics; and BG3 = less than 87.5% HF genetics). ssGBLUP = single-step genomic best linear unbiased prediction; wssGBLUP = weighted genomic best linear unbiased prediction.

**Table 5 animals-15-01320-t005:** Significant SNPs associated with MY305 in Thai-Holstein crossbred cattle.

No.	SNP	BTA	Location (bp)	SNP Variance	Gene	Size (bp)	Distance (bp)
1	ARS-BFGL-NGS-89632	4	13,968,183	1.12	*LOC618297*	37,202	on target
2	Hapmap59633-rs29013953	4	14,021,218	1.12	*SHFM1*	25,097	19,833
3	Hapmap45207-BTA-71975	4	14,104,597	0.93	*LOC101906304*	62,213	on target
4	Hapmap47403-BTA-76048	6	45,153,190	0.92	*PPARGC1A*	736,785	on target
5	UA-IFASA-3974	13	53,618,942	1.02	*PDYN*	19,519	on target
6	BTA-115023-no-rs	13	53,657,996	1.02	*SIRPA*	41,829	−10,958
7	ARS-BFGL-NGS-6380	18	57,788,407	1.23	*SIGLECL1*	4240	34,086
					*IGLON5*	14,439	10,978
					*VSIG10L*	16,522	on target
					*ETFB*	12,571	−7349
					*NKG7*	2760	−24,115
					*CLDND2*	1283	−20,706
					*LIM2*	7145	−31,072
8	Hapmap54014-rs29018901	24	541,784	1.32	*LOC529511*	11,567	20,116
9	BTA-57495-no-rs	24	653,401	1.15	*PARD6G*	77,202	on target
					*ADNP2*	21,361	−26,510
10	ARS-BFGL-NGS-117233	25	757,572	0.93	*LMF1*	51,476	on target
11	ARS-BFGL-NGS-62237	25	813,455	2.35	*SOX8*	5025	21,045
12	ARS-BFGL-NGS-116934	25	840,143	2.35	*SSTR5*	7249	−11,548
					*TEKT4*	5993	−30,987
					*C1QTNF8*	1656	−25,313
13	ARS-BFGL-NGS-115164	25	930,509	2.31	*CACNA1H*	27,689	−27,522
14	ARS-BFGL-NGS-117981	25	947,823	0.93	*LOC786948*	1884	−39,564
					*TPSB2*	1727	−46,728

*LOC618297* = uncharacteristic gene; *SHFM1* = split hand/foot malformation type 1; *LOC101906304* = uncharacteristic gene; *PPARGC1A* = PPARG coactivator 1 alpha; *PDYN* = prodynorphin; *SIRPA* = signal regulatory protein alpha; *SIGLECL1* = SIGLEC family like 1; *IGLON5* = IgLON family member 5; *VSIG10L* = v-set and immunoglobulin domain containing 10 like; *ETFB* = electron transfer flavoprotein subunit beta; *NKG7* = natural killer cell granule protein 7; *CLDND2* = claudin domain containing 2; *LIM2* = lens intrinsic membrane protein 2; *LOC529511* = uncharacteristic gene; *PARD6G* = par-6 family cell polarity regulator gamma; *ADNP2* = ADNP homeobox 2; *LMF1* = lipase maturation factor 1; *SOX8* = SRY (sex determining region Y)-box 8; *SSTR5* = somatostatin receptor 5; *TEKT4* = tektin 4; *C1QTNF8* = C1q and TNF related 8; *CACNA1H* = calcium voltage-gated channel subunit alpha1 H; *LOC786948* = uncharacteristic gene; *TPSB2* = tryptase beta 2.

**Table 6 animals-15-01320-t006:** Significant SNPs associated with DO in Thai-Holstein crossbred cattle.

No.	SNP	BTA	Location (bp)	SNP Variance	Gene	Size (bp)	Distance (bp)
1	UA-IFASA-7600	2	106,630,122	0.86	*TNS1*	146,152	on target
2	BTB-01610214	5	110,275,795	0.86	*MICALL1*	27,629	35,361
					*C5H22orf23*	7940	26,602
					*SOX10*	19,591	on target
					*POLR2F*	8273	18,274
3	ARS-BFGL-NGS-61240	5	110,300,207	0.93	*LOC101905166*	19,375	on target
4	ARS-BFGL-NGS-32908	5	110,320,607	0.87	*PICK1*	17,396	−25,546
					*SLC16A8*	7544	−44,410
5	ARS-BFGL-NGS-101279	5	110,399,710	0.87	*BAIAP2L2*	29,728	on target
					*PLA2G6*	58,871	−4115
6	UA-IFASA-3974	13	53,618,942	1.13	*PDYN*	19,519	on target
7	BTA-115023-no-rs	13	53,657,996	1.13	*SIRPA*	41,829	−10,958
8	ARS-BFGL-NGS-14448	13	53,766,894	0.86	*LOC107133069*	10,067	−17,866
					*LOC101902869*	14,890	−49,792
9	ARS-BFGL-NGS-109285	18	57,589,121	0.82	*LOC100138951*	9533	5484
					*LOC618463*	6560	on target
					*LOC100141212*	10,140	−18,876
10	ARS-BFGL-NGS-6380	18	57,788,407	1.06	*SIGLECL1*	4240	34,086
					*IGLON5*	14,439	10,978
					*VSIG10L*	16,522	on target
					*ETFB*	12,571	−7349
					*NKG7*	2760	−24,115
					*CLDND2*	1283	−20,706
					*LIM2*	7145	−31,072
11	ARS-BFGL-NGS-117233	25	757,572	0.80	*LMF1*	51,476	on target
12	ARS-BFGL-NGS-62237	25	813,455	1.95	*SOX8*	5025	21,045
13	ARS-BFGL-NGS-116934	25	840,143	1.95	*SSTR5*	7249	−11,548
					*TEKT4*	5993	−30,987
					*C1QTNF8*	1656	−25,313
14	ARS-BFGL-NGS-115164	25	930,509	1.89	*CACNA1H*	27,689	−27,522
15	ARS-BFGL-NGS-117981	25	947,823	0.88	*LOC786948*	1884	−39,564
					*TPSB2*	1727	−46,728
16	ARS-BFGL-NGS-6982	25	27,736,138	0.86	*ITGAD*	27,824	on target
					*COX6A2*	652	−1444
					*ARMC5*	8266	−14,085
					*LOC101905312*	3456	−11,058
					*TGFB1I1*	6787	−24,611
17	ARS-BFGL-NGS-70606	25	27,787,134	0.86	*SLC5A2*	9768	5761
					*RUSF1*	7231	−1137
					*AHSP*	1260	−13,671
18	Hapmap39031-BTA-59776	25	27,847,889	0.86	*LOC783313*	969	35,012
					*OR7D4*	939	9910
19	Hapmap44260-BTA-59780	25	27,872,006	0.87	*SEPT14*	34,793	on target
					*ZNF713*	19,695	−3663
					*MRPS17*	4268	−39,024

*TNS1* = tensin 1; *MICALL1* = MICAL like 1; *C5H22orf23* = chromosome 5 C22orf23 homolog; *SOX10* = sex determining region Y; *POLR2F* = RNA polymerase II; I; and III subunit F; *LOC101905166* = uncharacteristic gene; *PICK1* = protein interacting with C kinase 1; *SLC16A8* = solute carrier family 16 member 8; *BAIAP2L2* = BAR/IMD domain containing adaptor protein 2 like 2; *PLA2G6* = phospholipase A2 group VI; *PDYN* = prodynorphin; *SIRPA* = signal regulatory protein alpha; *LOC107133069* = uncharacteristic gene; *LOC101902869* = uncharacteristic gene; *LOC101902869* = uncharacteristic gene; *LOC100138951* = uncharacteristic gene; *LOC618463* = uncharacteristic gene; *LOC100141212* = uncharacteristic gene; *SIGLECL1* = SIGLEC family like 1; *IGLON5* = IgLON family member 5; *VSIG10L* = v-set and immunoglobulin domain containing 10 like; *ETFB* = electron transfer flavoprotein subunit beta; *NKG7* = natural killer cell granule protein 7; *CLDND2* = claudin domain containing 2; *LIM2* = lens intrinsic membrane protein 2; *LMF1* = lipase maturation factor 1; *SOX8* = SRY (sex determining region Y)-box 8; *SSTR5* = somatostatin receptor 5; *TEKT4* = tektin 4; *C1QTNF8* = C1q and TNF related 8; *CACNA1H* = calcium voltage-gated channel subunit alpha1 H; *LOC786948* = uncharacteristic gene; *TPSB2* = tryptase beta 2; *ITGAD* = integrin subunit alpha D; *COX6A2* = cytochrome c oxidase subunit 6A2; *ARMC5* = armadillo repeat containing 5; *LOC101905312* = uncharacteristic gene; *TGFB1I1* = transforming growth factor beta 1 induced transcript 1; *SLC5A2* = solute carrier family 5 member 2; *RUSF1* = RUS family member 1; *AHSP* = alpha hemoglobin stabilizing protein; *LOC783313* = uncharacteristic gene; *OR7D4* = olfactory receptor family 7 subfamily D member 4; *SEPT14* = septin-14; *ZNF713* = zinc finger protein 713; *MRPS17* = mitochondrial ribosomal protein S17.

**Table 7 animals-15-01320-t007:** Significant SNPs associated with milk FPR in Thai-Holstein crossbred cattle.

No.	SNP	BTA	Location (bp)	SNP Variance	Gene	Size (bp)	Distance (bp)
1	BTB-01610214	5	110,275,795	1.38	*MICALL1*	27,629	35,361
					*C5H22orf23*	7940	26,602
					*SOX10*	19,591	on target
					*POLR2F*	8273	18,274
2	ARS-BFGL-NGS-61240	5	110,300,207	1.50	*LOC101905166*	19,375	on target
3	ARS-BFGL-NGS-32908	5	110,320,607	1.40	*PICK1*	17,396	−25,546
					*SLC16A8*	7544	−44,410
4	ARS-BFGL-NGS-101279	5	110,399,710	1.40	*BAIAP2L2*	29,728	on target
					*PLA2G6*	58,871	−4115
5	ARS-BFGL-NGS-109285	18	57,589,121	1.13	*LOC100138951*	9533	5484
					*LOC618463*	6560	on target
					*LOC100141212*	10,140	−18,876
6	ARS-BFGL-NGS-6380	18	57,788,407	1.52	*SIGLECL1*	4240	34,086
					*IGLON5*	14,439	10,978
					*VSIG10L*	16,522	on target
					*ETFB*	12,571	−7349
					*NKG7*	2760	−24,115
					*CLDND2*	1283	−20,706
					*LIM2*	7145	−31,072
7	ARS-BFGL-NGS-79637	19	55,421,399	1.39	*LOC100196898*	9347	on target
8	ARS-BFGL-NGS-112332	19	55,479,408	1.39	*MGAT5B*	65,664	−48,354
9	ARS-BFGL-NGS-111549	19	55,548,825	0.99	*MGAT5B*	65,664	on target
10	ARS-BFGL-NGS-114189	19	55,590,126	0.98	*MGAT5B*	65,664	on target
					*MFSD11*	28,249	−47,373
11	ARS-BFGL-NGS-35280	21	33,475,697	1.43	*LINGO1*	217,647	on target
12	Hapmap40367-BTA-73682	21	33,515,083	1.44	*ODF3L1*	4209	−44,651
13	ARS-BFGL-NGS-62237	25	813,455	1.34	*SOX8*	5025	21,045
14	ARS-BFGL-NGS-116934	25	840,143	1.34	*SSTR5*	7249	−11,548
					*TEKT4*	5993	−30,987
					*C1QTNF8*	1656	−25,313
15	ARS-BFGL-NGS-115164	25	930,509	1.32	*CACNA1H*	27,689	−27,522
16	ARS-BFGL-NGS-117981	25	947,823	0.83	*LOC786948*	1884	−39,564
					*TPSB2*	1727	−46,728

*MICALL1* = MICAL like 1; *C5H22orf23* = chromosome 5 C22orf23 homolog; *SOX10* = sex determining region Y; *POLR2F* = RNA polymerase II; I; and III subunit F; *LOC101905166* = uncharacteristic gene; *PICK1* = protein interacting with C kinase 1; *SLC16A8* = solute carrier family 16 member 8; *BAIAP2L2* = BAR/IMD domain containing adaptor protein 2 like 2; *PLA2G6* = phospholipase A2 group VI; *LOC100138951* = uncharacteristic gene; *LOC618463* = uncharacteristic gene; *LOC100141212* = uncharacteristic gene; *SIGLECL1* = SIGLEC family like 1; *IGLON5* = IgLON family member 5; *VSIG10L* = v-set and immunoglobulin domain containing 10 like; *ETFB* = electron transfer flavoprotein subunit beta; *NKG7* = natural killer cell granule protein 7; *CLDND2* = claudin domain containing 2; *LIM2* = lens intrinsic membrane protein 2; LOC100196898; *MGAT5B* = alpha-1;6-mannosylglycoprotein 6-beta-N-acetylglucosaminyltransferase B; *MFSD11* = major facilitator superfamily domain containing 11; *LINGO1* = leucine rich repeat and Ig domain containing 1; *ODF3L1* = outer dense fiber of sperm tails 3 like 1; *SOX8* = SRY (sex determining region Y)-box 8; *SSTR5* = somatostatin receptor 5; *TEKT4* = tektin 4; *C1QTNF8* = C1q and TNF related 8; *CACNA1H* = calcium voltage-gated channel subunit alpha1 H; *LOC786948* = uncharacteristic gene; *TPSB2* = tryptase beta 2.

## Data Availability

Additional data are available upon request from the corresponding author.

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
