# Peer review of "Pleiotropic Genes Affecting Milk Production, Fertility, and Health in Thai-Holstein Crossbred Dairy Cattle: A GWAS Approach"

_animals, 2025, doi:10.3390/ani15091320_

Round 1
Reviewer 1 Report
Comments and Suggestions for Authors
Author deal with nowadays clasical design GWAS analysis of milk recording data with restricted number of individuals with genomic information and based on (weighted) single step GBLUP estimate basic genetic parameters. As a results of GWAS on several phenotypic data they provide information about genetic signals common.
I wa ssurprised that some of references in Introduction are bit "outdated" 20 years old, however still valid in general terms.
First point is that I miss the Aim of the study? Sorry but it essential to state the aim and possible hypotheses.
M&M
L. 95 2.1 Data collection?
this is critical part of this study, thre is unclear how 48274 first calvers could deliver in 8 years of collection only 18843 MY305 lactations? How than you obtained 48274 DO records? In clasical reproduction biology one DO preceeds one lactation....
When looking in to table 1 even more such questions arive:
47512 animals with records? what kind of records? clarify!
87958 animals with pedigree? What was the pedigree completeness than? ie those animals are in pedigree of those 18000 or those 48000 animals?
HYS effect 30299 c'mon 8 years x 12 months how many herds were included? what is the average herd size? Rest of methodology is description of mixed model equations and well known expressions on how to calculate heritability and correlations but I like this clasical style of methodology ... .
Coming to the results section: estimate of genetic parameters are generaly in the range to be found in literature, ie in case of correlations depending on the entire data variability which is in case of restricted size usually very low. This also works for the observed accuracy of the estimations. Authors proved that wssGBLUP performed better in this case. LAter part of results and subsequent extensive discussion are from my point of viw too much. To confirm pleiotropic effect authors should just follow the main line and not broadening the submission with those extensive referencing whih is decreasing overal message they wanted to disseminate.
Author Response
Respond to Reviewer 1
Author deal with nowadays clasical design GWAS analysis of milk recording data with restricted number of individuals with genomic information and based on (weighted) single step GBLUP estimate basic genetic parameters. As a results of GWAS on several phenotypic data they provide information about genetic signals common.
Respond:
Thank you for your thoughtful summary of our approach. You're right—we used a more classical GWAS design with weighted single-step GBLUP to make the best use of our data, especially given the limited number of genotyped animals. Our main goal was to estimate reliable genetic parameters and explore common genetic signals across the traits we studied. We're glad the design and intent came through clearly, and we appreciate your recognition of the approach. We have revised the manuscript according to your comments, with all changes highlighted in grey. We sincerely hope the updated version meets your expectations. Below are our detailed responses to each of your comments.
Comments:
- First point is that I miss the Aim of the study? Sorry but it essential to state the aim and possible hypotheses.
Respond 1: We added the a9m of this study in the background section. Please see lines 97 – 101.
- 95 2.1 Data collection?
This is critical part of this study, there is unclear how 48274 first calvers could deliver in 8 years of collection only 18843 MY305 lactations? How than you obtained 48274 DO records? In classical reproduction biology one DO preceeds one lactation.
Respond 2: Thank you for raising this important point. In this analysis, milk production and reproductive data were recorded separately. The AI data were recorded from all farms that used DLD’s bull, however, only sampling farms were collected the test-day milk and milk composition data. Datasets were merged using livestock ID, but some animals in the lactation data were missing from the DO dataset. Consequently, 18,843 records of first lactation MY305 and FPR were available, along with 48,274 first-calving DO. However, genomic analysis (ssGBLUP) could efficiently handle unbalanced data. We write an additional sentences for clarity. …Please see lines 107 – 109.
- When looking in to table 1 even more such questions arive:
47512 animals with records? what kind of records? clarify!
87958 animals with pedigree? What was the pedigree completeness than? ie those animals are in pedigree of those 18000 or those 48000 animals?
Respond 3: Thank you for the related questions. The 47,512 animals listed as having records refer to those with at least one phenotypic record whether for MY305, DO, or milk FPR. However, we deleted this line from the table, avoiding confusion. …Please see the line “animals with records” was removed in table 1.
The 87,958 animals with pedigree information include not only the animals with phenotypic records but also their parents and other related individuals, as obtained after running the renumbering process using the renumf90.exe program. This software generates pedigree for both animals with records either MY305 or DO, and their ancestors. Many of the animals in the pedigree do not have performance records themselves, but they are included to improve the accuracy of genetic evaluations through pedigree relationships.
- HYS effect 30299 c'mon 8 years x 12 months how many herds were included? what is the average herd size? Rest of methodology is description of mixed model equations and well known expressions on how to calculate heritability and correlations but I like this classical style of methodology.
Respond 4: The data were prepared by running the renumbering process using renumf90.exe software. The HMY effect reflects a combination of herd, month of calving, and year of calving for animals with available records.
We really appreciate your kind words about the classical approach in our methodology. Our goal was to keep things clear and well-structured, using methods that are widely accepted in animal breeding research. We added the details of data management in M&M part. …Please see lines 110 – 118.
- Coming to the results section: estimate of genetic parameters are generally in the range to be found in literature, ie in case of correlations depending on the entire data variability which is in case of restricted size usually very low. This also works for the observed accuracy of the estimations. Authors proved that wssGBLUP performed better in this case. Later part of results and subsequent extensive discussion are from my point of view too much. To confirm pleiotropic effect authors should just follow the main line and not broadening the submission with those extensive referencing which is decreasing overall message they wanted to disseminate.
Respond 5: Thank you for your feedback We're also glad to hear that our findings on the improved performance of wssGBLUP came through clearly. Regarding your comment on the later sections of the results and discussion. We made a revision the later part of discussion and try to focus on the main findings to confirm the pleiotropic genes. … Please see lines 642-677.
Reviewer 2 Report
Comments and Suggestions for Authors
In general, manuscript is well written and presents the results of elucidating the genetic mechanisms that underlie such complex traits milk production, fertility, and health in Thai-Holstein crossbred dairy cattle.
Comments
I recommend indicating the breeds, which were used in creating of Thai-Holstein crossbred cattle.
The crossbred dairy cows collected between 1999 and 2017 were investigated in this study. The data collection is represented by wide time window. The systems of feeding, keeping and management could be changed significantly. It is not clear how the authors addressed these differences (biases) in their models. This point should be clarified.
The network of gene connections will be useful for better presentation of the results.
Author Response
Respond to Reviewer 2
General comments: manuscript is well written and presents the results of elucidating the genetic mechanisms that underlie such complex traits milk production, fertility, and health in Thai-Holstein crossbred dairy cattle.
Thank you very much for your kind words and positive feedback. We're glad to hear that the manuscript was clear and well-written, and that the aims of uncovering the genetic mechanisms behind milk production, fertility, and health traits in Thai-Holstein crossbred cattle came through effectively. We have revised the manuscript according to your comments, with all changes highlighted in blue. We sincerely hope the updated manuscript will live up to your expectations. Below are our answers to each comment.
Comments
- I recommend indicating the breeds, which were used in creating of Thai-Holstein crossbred cattle.
Respond 1: we have added information on the breeds involved in the development of Thai-Holstein crossbred cattle. Specifically, these crossbreds were created by mating Holstein Friesians with Thai Native (Bos indicus) breeds. …Please see line 66-69.
- The crossbred dairy cows collected between 1999 and 2017 were investigated in this study. The data collection is represented by wide time window. The systems of feeding, keeping and management could be changed significantly. It is not clear how the authors addressed these differences (biases) in their models. This point should be clarified.
Respond 2: We realize that using data collected over such a long period (1999–2017) means there could have been changes in feeding, management, and overall farm practices. To help account for these differences, we included contemporary group effects of herd, year, and month of calving in our models. This allowed us to control for herd management variation over time across farms, Additionally, the management of Thai dairy cattle is controlled by the Department of Livestock Development (DLD), which helps standardize practices and further reduce environmental variation. HYM of calving as contemporary management group was used in the model analyses. Please see line 146-147.
- The network of gene connections will be useful for better presentation of the results.
Respond 2: We agree that gene network analysis can provide deep insights. After we have tried the analysis, we found that it is too complicated to explain about the pleiotropic genes which is the main objectives of this study. Therefore, we chose Venn’s diagram a in Figure 4.
Reviewer 3 Report
Comments and Suggestions for Authors
General comments:
This study aimed to identify pleiotropic genes associated with three parameters related to milk production and reproduction in Thai-Holstein crossbred dairy cattle. This subject is relevant. The study gives new insights toward genomic selection with particular emphasis to Thai-Holstein crossbred and improve knowledge about pleiotropic genes. Overall, the study is well done and presents a strong scientific soundness. The aim is lacking at final of the introduction section. A clarification of cows´ selection for genotyping should be made. Overall, results are well presented as supported by informative table and figures. Nonetheless some discussion was inserted in this section. The findings are adequately discussed and supported by relevant literature accurately inserted in the text. Conclusions are supported by results.
Specific comments:
L41: (DO-milk-FPR).
L91: Please add the aim of this study.
L103: I suggest reporting how the selection of these 882 was made.
227-229: This is part of discussion. Please check the results section for this issue. I suggest moving them for the discussion section.
L340-348: Can you quantify the correlations into the graphs of Figures 5-7?
L418: I suggest changing “tend” to other term.
L445: In our study…
Author Response
Respond to Reviewer 3
General comments:
This study aimed to identify pleiotropic genes associated with three parameters related to milk production and reproduction in Thai-Holstein crossbred dairy cattle. This subject is relevant. The study gives new insights toward genomic selection with particular emphasis to Thai-Holstein crossbred and improve knowledge about pleiotropic genes. Overall, the study is well done and presents a strong scientific soundness. The aim is lacking at final of the introduction section. A clarification of cows´ selection for genotyping should be made. Overall, results are well presented as supported by informative table and figures. Nonetheless some discussion was inserted in this section. The findings are adequately discussed and supported by relevant literature accurately inserted in the text. Conclusions are supported by results.
Thank you so much for your thoughtful and encouraging feedback. We're really glad to hear that you found the study relevant and scientifically sound, and that the tables and figures helped support the results clearly. We appreciate your suggestion about clarifying the study’s aim at the end of the introduction—we’ve revised that section to make the objective more explicit. Lastly, we went through the results section to clean up any parts that felt more like discussion and made sure everything is clearly organized. Your comments have been really helpful in strengthening the manuscript, and we truly appreciate your time and insight. We have revised the manuscript according to your comments and marked it in green. We sincerely hope the updated manuscript will live up to your expectations. Below are our answers to each comment.
Specific comments:
- L41: (DO-milk-FPR)
Respond 1: we have changed it into (DO and milk FPR).…Please see line 46.
- L91: Please add the aim of this study.
Respond 2: We added the aim of this study in the background section. …Please see lines 99 – 101.
- L103: I suggest reporting how the selection of these 882 was made.
Respond 3: Genotyped animals were randomly selected based on high, medium and low breeding values of bulls and dams. We added the selection criteria for genotyped animals as suggestion. …Please see lines 121-123.
- 227-229: This is part of discussion. Please check the results section for this issue. I suggest moving them for the discussion section.
Respond 4: We decided to remove and include them instead in the discussion section
- L340-348: Can you quantify the correlations into the graphs of Figures 5-7?
Respond 5: Yes, we can. We added correlation coefficients, and p-value in the Figure descriptions. …Please see Figure 5-7 (lines 361, 362, 368, 369, 375, 376)
- L418: I suggest changing “tend” to other term.
Respond 6: We changed it into often. … cows with higher MY305 often have lower milk FPR. …Please see the line 440
- L445: In our study…
Respond 7: we have added it into the sentence. …Please see the line 467.
Reviewer 4 Report
Comments and Suggestions for Authors
Fathoni et al perform GWAS and genomic prediction for three phenotypes (305-day milk yield (MY305), days open (DO), and milk fat-to-protein ratio (FPR) ) in Thai-Holstein crossbred dairy cattle.
Overall, the manuscript showed several novel results and might be important for genetic improvements of traits in this breed.
Below are some comments:
To improve the accuracy compared to what?
The simple summary in plain text, but should be informative
Line 26: What did the authors mean by the health traits here since the authors have three traits, please mention them in the simple summary.
Line 33: Add sample size and numbers of SNPs before and after QC.
Line 36-37: 5% of Genetic variance or phenotypic variance?
Line 37-41: Which type of heritability here, from the genomics or additive genetics (animal models)?
What are the models for computing heritability and what is the SE of h2?
Line 41: There is no clear evidence for trade-off; what did the authors mean?
How did the authors confirm assumptions about the pleiotropic effect (line 43)?
The introduction is fine but might extend more to finding the candidate genes for the traits in pure and crossbreed, especially when reported in tropical countries.
Line 101:102: What time of the day and periods during the lactation cycle do the authors measure the fat and protein percentage to derive the FPR? Why did the authors report the fat and protein percentages separately as they are important for milk quality as well?
Ratio trait is not an idea for selection.
Line 109-113: Did the authors perform the test to see any differences in the phenotypic (genetic) information between the three breed-based groups? How did the authors select the percentage to group the cows?
Line 149: Why did the authors consider it is weighted single step GBUP and GWAS, what is the weight factor here and how did the author derive the weight factor?
What is di in line 162?
Could the authors explore more about how to get the prediction accuracy of genomic predictions, and what is the training and validating populations here?
Did the authors check the population structure of the genotyped animals?
Table 4: The authors might provide more information about bulls and dams
Surprisingly, the authors did not report any SNPs in BTA14 for MY305. DGAT1 has been reported in many studies and significantly contributes to milk yield. The authors might discuss it.
Comments on the Quality of English LanguageOverall, the quality of English is fine.
Author Response
Respond to Reviewer 4
Thank you for your suggestions to improve our manuscript. We have revised it and marked it in yellow. We sincerely hope the updated manuscript will live up to your expectations. Below are our answers to each comment.
- Simple summary: To improve the accuracy compared to what?
The simple summary in plain text, but should be informative
Respond 1: compared to the standard single-step GBLUP. Please see lines 25 – 26
- Line 26: What did the authors mean by the health traits here since the authors have three traits, please mention them in the simple summary.
Respond 2: its mean fat-to-protein ration, we have mentioned them in the summary. Please see line 27
- Line 33: Add sample size and numbers of SNPs before and after QC.
Respond 3: We have added sample size and numbers of SNPs before and after QC in the abstract. Please see lines 34-37
- Line 36-37: 5% of Genetic variance or phenotypic variance?
Respond 4: 5% of the total genetic variance …Please see line 40-41.
- Line 37-41: Which type of heritability here, from the genomics or additive genetics (animal models)?.
Respond 5: the type of heritability was additive genetics …Please see line 43.
- What are the models for computing heritability and what is the SE of h2?
Respond 6: we used single-step genomic REML (ssGREML) with multi-trait animal model. SE is standard error …Please see line 41-42, 139-140.
- Line 41: There is no clear evidence for trade-off; what did the authors mean?
Respond 7: We change the word “trade-off” to “compromise”. … Please see line 47.
- How did the authors confirm assumptions about the pleiotropic effect (line 43)?
Respond 8: Pleiotropic effect was by SNPs showing significant influences on more than one trait, confirmed by the strong. … Please see lines 49-50.
- Line 101:102: What time of the day and periods during the lactation cycle do the authors measure the fat and protein percentage to derive the FPR? Why did the authors report the fat and protein percentages separately as they are important for milk quality as well? Ratio trait is not an idea for selection.
Respond 9: we calculated milk FPR for each test-day record and use the average value over lactation …Please see lines 115-116.
The milk FPR can serve as secondary indicator to determine negative energy balanced (NEB) and metabolic acidosis, which affect to milk yield and fertility traits. Milk FPR has been widely used in selection program. (For example: Negussie, E.; Strandén, I.; Mäntysaari, E. A. Genetic Associations of Test-Day Fat:Protein Ratio with Milk Yield, Fertility, and Udder Health Traits in Nordic Red Cattle. Journal of Dairy Science, 2013, 96 (2), 1237–1250. https://doi.org/10.3168/jds.2012-5720.)
- Line 109-113: Did the authors perform the test to see any differences in the phenotypic (genetic) information between the three breed-based groups? How did the authors select the percentage to group the cows?
- Respond 10: Testing among Holstein-based groups has been done in several previous reports. Increasing Holstein genetics tends to increase milk yield. Higher milk yield is associated with decreased fertility, which consequently increases the number of days open. The percentage groups are defined by DLD’s dairy committee of Thailand. So, we followed the DLD suggestion.
- Line 149: Why did the authors consider it is weighted single step GBUP and GWAS, what is the weight factor here and how did the author derive the weight factor?
Respond 11: wssGBLUP and wssGWAS are advanced methods for genomic prediction and genome-wide association analysis which previously reported by other researchers. In this study, we applied these methods to our population of tropical dairy cattle. The details of weight were explained in materials and method section. …See lines 177-183.
- What is in line 162?
Respond 12: represents the weight for the th SNP, the formula for di was shown in line 179.
- Could the authors explore more about how to get the prediction accuracy of genomic predictions, and what is the training and validating populations here?
Respond 13: In our study, accuracy was estimated from prediction error variance (PEV), not by cross-validation so no training and validating population used. The PEV or ACC formula is commonly known, also available within the BLUPF90+ package. However, we give details of the accuracy in the materials and methods (…Please see lines 185-187) and change the word “prediction accuracy” to “GEBV accuracy” through the manuscript.
- Did the authors check the population structure of the genotyped animals?
Respond 14: Genotyped animals were randomly selected based on high, medium and low breeding values of bulls and dams, therefore the genotyped animals in our study were distributed from whole population. …Please see lines 121-123.
- Table 4: The authors might provide more information about bulls and dams
Respond 15: we provided more information about bulls and dams. …Please see lines 275-278.
- Surprisingly, the authors did not report any SNPs in BTA14 for MY305. DGAT1 has been reported in many studies and significantly contributes to milk yield. The authors might discuss it.
Respond 16: Thank you for your concern. There has been debate about DGAT1 effects on milk yield and %fat in crossbred dairy cows, particularly with zebu breeds in tropical countries. Non-significant of DGAT1 have been reported in the Thailand population, which might be due to factors such as low production, low quality of roughages, and heat stress. In this study no significant SNPs on BTA14 related to MY305 were found.